# The influence of body composition and fat distribution on circadian blood pressure rhythm and nocturnal mean arterial pressure dipping in patients with obesity

**Marek Tałałaj**[1]*, **Agata Bogołowska-Stieblich**[2], **Michał Wąsowski**[2], **Ada Sawicka**[2], **Piotr Jankowski**[2]

1 Department of Orthopedics, Pediatric Orthopedics and Traumatology, Centre of Postgraduate Medical Education, Warsaw, Poland, 2 Department of Internal Medicine and Geriatric Cardiology, Centre of Postgraduate Medical Education, Warsaw, Poland

* mtalalaj@cmkp.edu.pl

## Abstract

Loss of physiological nocturnal blood pressure (BP) decline is an independent predictor of cardiovascular risk and mortality. The aim of the study was to investigate the influence of body composition and fat distribution on 24-hour BP pattern and nocturnal dipping of mean arterial pressure (MAP) in patients with obesity. The study comprised 436 patients, 18 to 65 years old (306 women), with BMI $\geq$30 kg/m$^2$. Body composition was assessed with dual-energy X-ray absorptiometry (DXA) and blood pressure was assessed by 24-hour BP monitoring. The prevalence of hypertension was 64.5% in patients with BMI <40 kg/m$^2$ and increased to 78.7% in individuals with BMI $\geq$50 kg/m$^2$ (p = 0.034). The whole-body DXA scans showed that the hypertensive patients were characterized by a greater lean body mass (LBM) and a higher abdominal-fat-to-total-fat-mass ratio (AbdF/FM), while the normotensive participants had greater fat mass, higher body fat percentage and more peripheral fat. Loss of physiological nocturnal MAP decline was diagnosed in 50.2% of the patients. The percentage of non-dippers increased significantly: from 38.2% in patients with BMI <40 kg/m$^2$ to 50.3% in those with BMI 40.0–44.9 kg/m$^2$, 59.0% in patients with BMI 45.0–49.9 kg/m$^2$, 71.4% in those with BMI 50.0–54.9 kg/m$^2$ and 83.3% in patients with BMI $\geq$55 kg/m$^2$ (p = 0.032, p = 0.003, p<0.001, and p = 0.002 vs. BMI <40 kg/m$^2$, respectively). The multi-variable regression analysis showed that patients at the highest quartiles of body weight, BMI, LBM and AbdF/FM had significantly reduced nocturnal MAP dipping compared with patients at the lowest quartiles, respectively.

## Introduction

The prevalence of obesity is increasing dramatically worldwide. If the current trend continues, by the year 2025, 18% of men and 21% of women will be obese [1]. The simplest and most common method for diagnosing obesity is to calculate the body mass index (BMI), but this

**Data Availability Statement:** All relevant data are within the manuscript and its Supporting Information files.

**Funding:** The study was supported by the Grant Number 501-3-40-10-15, Centre of Postgraduate Medical Education, Warsaw, Poland. The funders had no role in the study design, data collection and analysis, the decision to publish or the preparation of the manuscript.

**Competing interests:** The authors have decleared that no competing interests exist.

measure cannot differentiate between individuals with excess fat and those with high muscle mass [2]. Body composition and fat distribution can be assessed precisely by dual-energy X-ray absorptiometry (DXA), which is an established technique in the non-invasive assessment of body composition. Whole-body DXA scans offer accurate measures of three main body components: bone mineral content, fat mass and lean body mass. Moreover, they allow to determine the amount of centrally located (abdominal) fat, which is more significant than peripheral, subcutaneous adipose tissue in predicting the risk of cardiometabolic sequelae associated with obesity [3]. Visceral obesity appears to be especially important in activation of the sympathetic nervous system and the renin-angiotensin-aldosterone system, thereby increasing the risk of developing hypertension [4,5].

Twenty-four-hour ambulatory blood pressure monitoring (ABPM) is regarded as the best currently available method of measuring blood pressure (BP) and its circadian profile [6]. According to 2018 European Society of Cardiology guidelines, the ABPM threshold for a diagnosis of hypertension is $\geq$130/80 mm Hg, while a BP below those values with a nocturnal BP decline of 10% to 20% are considered normal values for ABPM [7]. The maintenance of a physiological circadian rhythm in blood pressure with a nocturnal "dip" is important for cardiovascular health. A loss of nocturnal BP decline (non-dipping) is an independent predictor of cardiovascular risk and mortality [8,9]. Moreover, it is associated with a higher risk of microalbuminuria, hypertensive retinopathy [10] and asymptomatic lacunar infarction of the brain [11].

Both, office BP measurements and ABPM, measure the peripheral blood pressure at the brachial artery. However, vital organs such as the brain, the heart and the kidneys are exposed to central (aortic) pressure; therefore, central systolic BP (SBP) is a better predictor of cardiovascular events than SBP measured at the upper arm [12,13]. Systolic BP amplifies from the aorta toward the peripheral arteries due to the arterial vessels' progressive reduction in diameter and increase in stiffness [14]. In young and middle-aged people the difference between central and peripheral SBP is approximately 10 mmHg [15,16], but this significantly increases with ageing [17]. Contrary to systolic BP, mean arterial pressure (MAP) remains relatively constant along the arterial tree, and it therefore better reflects central BP and shows a closer relationship with hypertension-associated organ damage [12,18].

This study aimed to investigate (1) the influence of body composition and fat distribution– as determined with DXA–on the 24-hour systolic and diastolic BP profiles in patients with obesity as well as (2) the associations of obesity and central obesity with a non-dipping pattern of MAP, as an indirect indicator of central SBP.

## Material and methods

A group of 436 Caucasian patients with obesity (306 women and 130 men) were enrolled in the study. The exclusion criteria comprised pregnancy, secondary or drug-induced obesity (e.g. due to endocrine disorders) or any physical disability that might interfere with the measurement of body composition (amputation or orthopedic prosthesis). Participants with a body weight >190 kg and/or a height >190 cm were also excluded due to the limitations of the densitometry device.

The patients were between the ages of 18 and 65 years and had BMI $\geq$30 kg/m$^2$. In all patients main anthropometric parameters were determined: body weight (BW) was measured to the nearest 0.1 kg with a digital scale, and standing height was measured to the nearest 0.1 cm with a fixed wall stadiometer. The weight and height were measured with the participants wearing light indoor clothing and no shoes. Waist and hip circumferences (WC, HC) were measured to the nearest 1 cm, and the waist-to-hip ratio (WHR) was calculated. WC was

determined in a horizontal plane, midway between the iliac crest and the costal margin, at the end of a normal expiration; HC was measured at the level of the greater trochanter, using a non-stretch tape measure [19].

Whole-body DXA scans were performed using HOLOGIC DELPHI system with QDR Software (v.11.1, Hologic, Bedford, MA). Automatic daily quality control of the system was performed with a spine phantom provided by the manufacturer. Using the DXA technique, lean body mass (LBM), whole body fat mass (FM), body fat percentage (BF%) and peripheral fat (PerF) were determined. Abdominal fat (AbdF) was measured within the region from the top of the pubic bone, extending cranially up to the line between twelfth thoracic and first lumbar vertebrae, as previously described [20]. After the scan was completed, the abdominal-fat-to-total-fat-mass ratio (AbdF/FM) was calculated.

Each patient underwent 24-hour BP monitoring with a BTL 08 CardioPoint ABP monitor (BTL Industries Ltd., Stevenage, UK). The measurements were performed according to the American Heart Association guidelines [6]. Systolic and diastolic BP (DBP) were measured every 30 minutes during the day (between 6.00 and 22.00) and every 60 minutes during the night (from 22.00 to 6.00). The average daytime and nighttime BP were computed as the means of all readings during each period. The average circadian SBP and DBP were calculated according to the formula (2 x mean daytime BP + mean nightime BP) / 3, that reflected the times of diurnal and nocturnal BP measurements, 16 hours and 8 hours, respectively, and eliminated the influence of different measurement frequencies on average 24-hour BP values. Hypertension was diagnosed according to the 2018 European Society of Cardiology guidelines [7]. Non-dipping of BP was defined as a decline in MAP values <10% from the average daytime to the average nighttime values.

The study was performed in accordance with the principles of the Declaration of Helsinki, and it was approved by the Ethics Committee of the Centre of Postgraduate Medical Education, Warsaw, Poland. Written informed consent was obtained from all participants.

### Statistical analysis

The results are presented as mean ± standard deviation (±SD). The normality of the continuous variables' distribution was verified with the Shapiro-Wilk test, and the homogeneity of the variance was determined with Levene's test. Paired or nonpaired Student's t-tests were performed to compare parametric data, while the Mann-Whitney U test was used to compare nonparametric variables. For the correlation assessment, the values of Spearman's coefficient (r) were computed. Categorical variables are expressed as numbers and percentages. Univariate comparisons were made between groups using the $\chi2$ test for categorical variables. Multivariable linear regression models were used to investigate associations between anthropometric parameters/body composition and MAP dipping. Independent variables were categorized according to the quartiles. Stepwise selection with Akaike's information criterion was performed to select variables for multivariable models. The calculations were performed using Statistica software, version 13.1. A p-value < 0.05 was considered statistically significant.

### Results

The mean age of the patients was 43 ± 11 years. Their basic anthropometric data, body composition and daytime, nighttime and average BP are presented in Table 1.

Out of 436 obese patients, 395 (90.6%) had BMI $\geq$35 kg/m$^2$, 284 (65.1%) suffered from morbid obesity with BMI $\geq$40 kg/m$^2$ and 47 (10.8%) had BMI $\geq$50 kg/m$^2$. Total of 291 patients (66.7%) were considered hypertensive. Up to 6 blood pressure-lowering drugs were used by 289 participants; the mean number of medications was 2.54 ±1.11. The prevalence of

**Table 1. Anthropometric parameters, body composition and blood pressure in patients with obesity.**

|  | Mean ± SD (range) | |
| --- | --- | --- |
| BW (kg) | 120.4 ± 20.2 | (73.3–189.0) |
| BMI (kg/m$^2$) | 42.4 ± 6.0 | (30.0–64.9) |
| WC (cm) | 123 ± 14 | (88–175) |
| HC (cm) | 128 ± 15 | (95–175) |
| WHR (cm/cm) | 0.96 ± 0.10 | (0.72–1.25) |
| LBM (kg) | 62.9 ± 11.5 | (38.0–100.9) |
| FM (kg) | 49.1 ± 11.0 | (22.3–83.1) |
| BF% | 42.7 ± 6.3 | (21.7–59.5) |
| PerF (kg) | 35.9 ± 9.8 | (11.6–64.9) |
| AbdF (kg) | 13.2 ± 2.9 | (5.6–24.2) |
| AbdF/FM (%) | 27.8 ± 6.4 | (9.7–52.1) |
| SBP—daytime (mmHg) | 131 ± 15 | (91–195) |
| DBP—daytime (mmHg) | 78 ± 8 | (58–106) |
| SBP—nighttime (mmHg) | 121 ± 17 | (87–198) |
| DBP—nighttime (mmHg) | 69 ± 9 | (47–109) |
| SBP—mean (mmHg) | 127 ± 15 | (90–194) |
| DBP—mean (mmHg) | 75 ± 8 | (58–102) |

BW, body weight; BMI, body mass index; WC, waist circumference; HC, hip circumference; WHR, waist-to-hip ratio; LBM, lean body mass; FM, fat mass; BF%, body fat percentage; PerF, peripheral fat; AbdF, abdominal fat; AbdF/FM, abdominal-fat-to-total-fat-mass ratio; SBP, systolic blood pressure; DBP, diastolic blood pressure.

hypertension increased together with BMI values, from 64.5% (BMI <40 kg/m$^2$) to 78.7% (BMI ≥50 kg/m$^2$) (Table 2).

Hypertension was found in 179 women (58.5%) and 112 men (86.2%) (p<0.001), despite the participants of both sexes being of similar age and having similar BMI values. The men were characterized by higher SBP and DBP than the women. Daytime SBP and DBP were 4.9% and 6.7% higher, respectively, while the nighttime values were 6.3% and 10.4% higher, respectively. Pulse pressure (PP) was similar in both sexes (Table 3).

Significant differences were found in body size and composition between the hypertensive and normotensive patients (Table 4). The former had 4.1% higher BW, 4.4% larger WC and 4.3% higher WHR, while the BMI and HC values were similar between the groups. The DXA scans showed that the hypertensive participants had 6.3% higher LBM and 13.3% higher AbdF/FM, while the normotensive individuals had 7.5% higher FM, 7.5% higher BF%, and 12.2% higher PerF. The extended analysis examining variables by sex are shown in S1–S3 Tables.

**Table 2. The number and percentage of hypertensive patients depending on BMI values.**

| BMI (kg/m$^2$) | All patients (n) | Hypertensive patients (n) | Hypertensive patients (%) | p-value |
| --- | --- | --- | --- | --- |
| <40 | 152 | 98 | 64.5 | |
| 40.0–44.9 | 159 | 102 | 64.2 | 0.478 |
| 45.0–49.9 | 78 | 54 | 69.2 | 0.224 |
| ≥50.0 | 47 | 37 | 78.7 | 0.034 |

BMI, body mass index; p-value vs. BMI <40 kg/m$^2$.

**Table 3. Age, BMI and blood pressure in female and male patients with obesity.**

|  | Women (n = 306) | Men (n = 130) | p-value |
|---|---|---|---|
| Age (years) | 42 ± 11 | 45 ± 11 | 0.057 |
| BMI (kg/m2) | 42.5 ± 5.8 | 42.1 ± 6.4 | 0.529 |
| SBP—daytime (mmHg) | 129 ± 15 | 135 ± 14 | < 0.001 |
| DBP—daytime (mmHg) | 76 ± 8 | 81 ± 8 | < 0.001 |
| PP—daytime (mmHg) | 53 ± 10 | 54 ± 9 | 0.247 |
| SBP—nighttime (mmHg) | 118 ± 17 | 126 ± 15 | 0.001 |
| DBP—nighttime (mmHg) | 67 ± 9 | 74 ± 10 | < 0.001 |
| PP—nighttime (mmHg) | 51 ± 11 | 52 ± 9 | 0.628 |
| SBP—mean (mmHg) | 125 ± 15 | 132 ± 14 | < 0.001 |
| DBP—mean (mmHg) | 73 ± 8 | 79 ± 8 | < 0.001 |
| PP—mean (mmHg) | 52 ± 10 | 53 ± 9 | 0.345 |

SBP, systolic blood pressure; DBP, diastolic blood pressure; PP, pulse pressure.

Data are expressed as mean ± SD.

Correlations between the anthropometric measures of obesity, body composition and blood pressure are presented in Table 5. It was found that BMI positively correlated with SBP and PP, while the other parameters related to body size—BW and LBM—positively correlated with both SBP and DBP, as well as with PP. The measures of central adiposity—WC, AbdF and AbdF/FM—positively correlated with SBP and DBP, but not with PP, whereas WHR positively correlated with DBP alone. Hip circumference, FM and PerF did not influence blood pressure, while BF% negatively correlated with both SBP and DBP values.

The assessment of nocturnal BP decline revealed that the magnitude of SBP dipping (9.9 ± 8.2 mmHg) was greater than that of the DBP dipping (8.5 ± 7.0 mmHg) by 1.4 ± 5.2 mmHg, p<0.001, while the percentage of the DBP decline (10.8 ± 8.7%) was greater than that of SBP decline (7.6 ± 6.8%) by 3.2 ± 4.9%, p<0.001. The average MAP dipping was 9.4 ± 7.5%. Reduced nocturnal MAP decline was diagnosed in 219 patients (50.2%). The percentage of non-dippers increased significantly along with higher BMI from 38.2% of participants with a BMI <40 kg/m$^2$ up to 83.3% of patients with a BMI ≥55 kg/m$^2$ (Table 6). The extended

**Table 4. Anthropometric parameters and body composition in hypertensive and normotensive patients with obesity.**

|  | Hypertensive patients (n = 291) | Normotensive patients (n = 145) | p-value |
|---|---|---|---|
| BW (kg) | 122.0 ± 21.9 | 117.2 ± 15.8 | 0.019 |
| BMI (kg/m$^2$) | 42.7 ± 6.4 | 41.7 ± 5.0 | 0.103 |
| WC (cm) | 125 ± 15 | 120 ± 13 | 0.005 |
| HC (cm) | 128 ± 16 | 128 ± 13 | 0.979 |
| WHR (cm/cm) | 0.98 ± 0.10 | 0.94 ± 0.09 | 0.001 |
| LBM (kg) | 64.3 ± 12.7 | 60.5 ± 8.7 | 0.004 |
| FM (kg) | 47.8 ± 11.3 | 51.4 ± 10.2 | 0.004 |
| BF% | 41.6 ± 6.6 | 44.7 ± 5.0 | 0.001 |
| PerF (kg) | 34.3 ± 9.9 | 38.5 ± 9.1 | 0.001 |
| AbdF (kg) | 13.4 ± 3.1 | 12.9 ± 2.4 | 0.093 |
| AbdF/FM (%) | 29.0 ± 6.8 | 25.6 ± 4.8 | < 0.001 |

BW, body weight; BMI, body mass index; WC, waist circumference; HC, hip circumference; WHR, waist-to-hip ratio; LBM, lean body mass; FM, fat mass; BF%, body fat percentage; PerF, peripheral fat; AbdF, abdominal fat; AbdF/FM, abdominal-fat-to-total-fat-mass ratio. Data are expressed as mean ± SD.

**Table 5. Correlations between anthropometric characteristics/body composition and blood pressure in patients with obesity.**

|  | Daytime | | | Nighttime | | | Average | | |
|---|---|---|---|---|---|---|---|---|---|
|  | SBP | DBP | PP | SBP | DBP | PP | SBP | DBP | PP |
| BW | 0.21*** | 0.18*** | 0.18*** | 0.28*** | 0.23*** | 0.21*** | 0.24*** | 0.21*** | 0.19*** |
| BMI | 0.15** | 0.06 | 0.16*** | 0.22*** | 0.10 | 0.22*** | 0.18*** | 0.08 | 0.18*** |
| WC | 0.12* | 0.17** | 0.06 | 0.22*** | 0.25*** | 0.10 | 0.16** | 0.21*** | 0.07 |
| HC | 0.07 | 0.02 | 0.09 | 0.10 | 0.05 | 0.10 | 0.09 | 0.02 | 0.10 |
| WHR | 0.07 | 0.19** | -0.03 | 0.10 | 0.22*** | -0.05 | 0.09 | 0.22*** | -0.04 |
| LBM | 0.20*** | 0.20*** | 0.14** | 0.25*** | 0.26*** | 0.13* | 0.22*** | 0.24*** | 0.13* |
| FM | 0.02 | 0.01 | 0.03 | 0.04 | -0.01 | 0.09 | 0.03 | 0.00 | 0.06 |
| BF% | -0.12* | -0.13* | -0.05 | -0.13* | -0.18** | 0.01 | -0.12* | -0.16** | -0.03 |
| PerF | -0.00 | -0.02 | 0.02 | 0.00 | -0.06 | 0.08 | -0.01 | -0.04 | 0.04 |
| AbdF | 0.15** | 0.16** | 0.10 | 0.23*** | 0.22*** | 0.11 | 0.18*** | 0.18*** | 0.10 |
| AbdF/FM | 0.13* | 0.15** | 0.07 | 0.18** | 0.20*** | 0.06 | 0.15** | 0.18*** | 0.06 |

BW, body weight; BMI, body mass index; WC, waist circumference; HC, hip circumference; WHR, waist-to-hip ratio; LBM, lean body mass; FM, fat mass; BF%, body fat percentage; PerF, peripheral fat; AbdF, abdominal fat; AbdF/FM, abdominal-fat-to-total-fat-mass ratio; SBP, systolic blood pressure; DBP, diastolic blood pressure; PP, pulse pressure.

*p < 0.05

**p < 0.01

***p < 0.001.

analysis of how hypertension status within BMI group relates to the nocturnal MAP dipping status is shown in S1 and S2 Figs.

The stepwise linear regression analysis (with BW, BMI, WC, WHR, LBM, FM, BF%, PerF, AbdF and AbdF/FM tested for inclusion in the multivariable models) revealed significant negative associations between BW, BMI, LBM, AbdF/F and nocturnal dipping of MAP (Table 7).

It was found that patients with BW ≥130 kg and those with BMI ≥45.5 kg/m$^2$ demonstrated over 3% lower nocturnal MAP dipping compared with patients in the lowest quartiles of BW and BMI (β -3.09, CI -5.84 ÷ -0.34 and β -3.51, CI -6.31 ÷ -0.72, respectively). It was also shown that participants in the highest quartiles of LBM (≥69.1 kg) and AbdF/FM (≥23,6%) showed a lower nocturnal MAP decrease, by 4.01% and by 3.81%, respectively (β -4.01, CI -6.73 ÷ -1.29 and β 3.81, CI -6.59 ÷ -1.04, respectively) compared with patients in the lowest quartiles of LBM and AbdF/FM.

## Discussion

Our study examined the complex associations between body composition and fat distribution, and circadian SBP, DBP and MAP profile in obese patients. We found that the prevalence of

**Table 6. The number and percentage of non-dippers by BMI values.**

| BMI (kg/m$^2$) | All patients (n) | Non- dippers (n) | Non- dippers (%) | p-value |
|---|---|---|---|---|
| <40 | 152 | 58 | 38.2 |  |
| 40.0–44.9 | 159 | 80 | 50.3 | 0.032 |
| 45.0–49.9 | 78 | 46 | 59.0 | 0.003 |
| 50.0–54.9 | 35 | 25 | 71.4 | < 0.001 |
| ≥55.0 | 12 | 10 | 83.3 | 0.002 |

BMI, body mass index; p-value vs. BMI <40 kg/m$^2$.

**Table 7. Multiple stepwise regression analysis showing associations between anthropometric parameters/body composition and nocturnal mean arterial pressure decline.**

| | MAP dipping [%] | |
|---|---|---|
| | β | 95% CI |
| BW (kg) Ref.: 73.3–102.9 | | |
| 103.0–116.6 | -0.33 | -3.08 ÷ 2.42 |
| 116.7–129.9 | 0.65 | -2.08 ÷ 3.39 |
| 130.0–180.0 | -3.09* | -5.84 ÷ -0.34 |
| BMI (kg/m$^2$) Ref.: 35.5–37.5 | | |
| 37.6–41.2 | -0.97 | -3.72 ÷ 1.78 |
| 41.3–45.4 | -1.70 | -4.44 ÷ 1.04 |
| 45.5–62.8 | -3.51* | -6.31 ÷ -0.72 |
| LBM (kg) Ref.: 38.0–54.4 | | |
| 54.5–61.1 | -1.54 | -4.26 ÷ 1.18 |
| 61.2–69.0 | -0.07 | -2.79 ÷ 2.65 |
| 69.1–100.9 | -4.01** | -6.73 ÷ -1.29 |
| AbdF/FM (%) Ref.: 9.7–23.5 | | |
| 23.6–26.6 | 0.88 | -1.82 ÷ 3.59 |
| 26.7–30.8 | 0.71 | -2.00 ÷ 3.43 |
| 30.9–50.3 | -3.81** | -6.59 ÷ -1.04 |

MAP, mean arterial pressure; BW, body weight; BMI, body mass index; LBM, lean body mass; AbdF/FM, abdominal-fat-to-total-fat-mass ratio.

* $p < 0.05$

** $p < 0.01$.

hypertension in participants with obesity was 66.7%, which significantly increased to 78.7% in patients with BMI ≥50 kg/m$^2$. In a German study that included over 45,000 primary care patients, the prevalence of hypertension was found to be 72.9% in grade 1 obesity, 77.1% in grade 2 and 74.1% in grade 3 obesity [21]. These values were more than twice as high as the global age-standardized prevalence of hypertension: 32% in women and 34% in men aged 30–79 years [22]. The authors of the Korea National Health and Nutrition Examination Survey showed that obesity was associated with even higher odds of hypertension (OR, 7.54; 95% CI, 5.89–9.65) as compared to BMI 18.5–23 kg/m$^2$ [23]. Among the participants of the second Nurses' Health Study, the incidence of hypertension in obese women was almost five times higher than in those with BMI <23.0 kg/m$^2$ [24]. In contrast, a cross-sectional study that comprised young participants aged 5–21 years showed that the prevalence of hypertension was not significantly different between lean individuals (40.3%) and obese participants (43.7%) [25].

Previous studies had documented the relationships between BP and anthropometric measures of central obesity, such as WC and WHR. It had been suggested that these parameters, especially when used in combination, might be a superior predictor of obesity-related cardiometabolic risk [26,27]. A study by Momin et al. found that, when compared with participants with normal WC, abdominal obesity was positively associated with the incidence of hypertension in both men (OR = 1.79, 95% CI: 1.10–2.91) and women (OR = 1.61, 95% CI: 1.09–2.40) [28]. A Chinese study on over 4,000 adult participants showed that hypertension was associated with central obesity (OR = 2.21, 95% CI: 1.90–2.57) [29].

Few studies have evaluated the relationship between BP and body composition. Lee et al. revealed that the percentage of whole body fat, measured with DXA, was positively associated with higher odds of hypertension, determined with a standard mercury sphygmomanometer

(OR, 3.56 for highest vs. lowest quartile) [23]. A study carried out among South African children found significant positive correlations between systolic BP and body fat mass, fat percentage and fat-free mass. However, body composition was estimated based on triceps, gluteal and subscapular skinfold measurements [30]. Goto et al. showed that the prevalence of hypertension was positively associated with both visceral and subcutaneous fat area, measured with computed tomography at the level of the umbilicus [31]. The opposite results were shown in the Dallas Heart Study, which included 2595 adult participants with a mean BMI 29 kg/m$^2$. In multivariable-adjusted models, more lower body subcutaneous fat—as measured with DXA—delineated by oblique lines crossing the femoral neck and including gluteal-femoral fat—was associated with lower BP values. In contrast, more visceral fat, as determined by magnetic resonance imaging, was associated with higher SBP [32].

A study on 8802 US residents who participated in the 1999–2004 US National Health and Nutrition Examination Survey (NHANES) investigated the association between regional fat distribution, measured with DXA, and cardiometabolic risk factors. In multivariate-adjusted models, leg fat accumulation was inversely associated with systolic and diastolic BP [33]. The results of our study resemble those reported in the NHANES and the Dallas Heart Study. We found that a higher amount of centrally located, abdominal fat was associated with increased BP, while peripheral adipose tissue seemed to play a protective role against hypertension. It should be noted that the measures of central adiposity correlated positively with SBP and DBP, but not with PP; BW, BMI and LBM, on the other hand, were associated with higher SBP and PP reflecting the stiffness of large arteries [34].

The circadian BP rhythm is essential for maintaining the body's normal physiological functions [11,35,36]. It was estimated that 25%-30% of the normal population exhibited insufficient nocturnal SBP decline, and approximately 10%-20% of the population had less than normal DBP dipping [37]. In overweight individuals with hypertension, the prevalence of physiological SBP decline was found to be 15% lower than in their lean counterparts [38]. The authors of a European study on 3216 adults referred to a hypertension clinic, found a non-dipping pattern of BP in 45% of normal-weight patients and 65% of obese participants. However, the association between BMI and nighttime BP was not statistically significant after multivariable adjustment [39]. Similar results were obtained in studies involving young populations. A study by Macumber et al. reported that 34.4% of obese children (≥95th BMI percentile) and 13.6% of children in the lean group (15th–85th BMI percentiles) were non-dippers. Compared to the lean subjects, obese children had an adjusted prevalence ratio for SBP non-dipping of 2.15 (95% CI: 1.25–3.42), but increasing severity of obesity was not further associated with nocturnal non-dipping [25]. A retrospective chart review conducted among 263 young study participants revealed that the prevalence of SBP non-dipping among the obese patients was 45%, which increased as the severity of obesity increased [40].

A study that examined the association between measures of central obesity and circadian BP profile in 104 adult patients, found that WC and waist-to-height ratio were significantly higher in non-dippers. It also found that the percentage of systolic and diastolic nocturnal drop was significantly correlated with waist-to-height ratio [41]. In contrast, the authors of a Korean study involving 1290 subjects found no association between central obesity (defined as having a waist circumference ≥90 cm in males and ≥ 85 cm in females) and nocturnal dipping patterns. The percentage of non-dippers was 46% in the non-obese group and 47% in the central obesity group. Nocturnal SBP decline was also similar among the non-obese (7.32 ± 8.64%) and obese (7.17 ± 8.62%) participants [42]. A Polish study on 206 young adults, aged 18-35 years with BMI 18.5-35.0 kg/m$^2$, assessed the association between visceral adiposity and SBP non-dipping pattern. Visceral adipose tissue, measured with DXA at the upper part of the abdomen, was a relatively small fat deposit, constituting only 1.5% of patient weight and

3.1% of total body fat. In multiple regression analysis, significant correlations were found between visceral fat-to-BW ratio and the percentage of SBP nocturnal dipping (β = −0.375; 95% CI: −0.7167 to −0.042; p = 0.015). In the young men (but not the women), excess visceral fat (≥0.993 kg) was found to increase the odds by 2.3 times for non-dipping SBP [43].

It has been documented that nighttime BP was a stronger predictor of total mortality and major cardiovascular events than daytime BP [44]. Boggia et al. pooled data from 6 population-based cohorts and reported that a higher SBP at night and a higher night-to-day SBP ratio were associated with a higher risk of all-cause and cardiovascular mortality, independent of mean awake and 24-hour SBP [45]. In children with pediatric-onset systemic lupus erythematosus, isolated nocturnal BP non-dipping was associated with endothelial dysfunction and atherosclerotic changes [46]. The MAPEC study found that subjects taking at least 1 antihypertensive medication at bedtime exhibited a lower cardiovascular risk than those who ingested all drugs in the morning [47].

To the best of our knowledge, the present study is the first to show the association between total body weight/composition and nocturnal decline in MAP. Previous studies have documented that peripheral SBP and DBP followed a similar nocturnal dipping profile to that of central BP, but MAP better reflected central, aortic SBP and better predicted cardiovascular risk than brachial SBP [15,48,49].

The studies that determined central BP through the acquisition of brachial waveform, with an automated oscillometric device, showed that waveform calibration with cuff-based MAP/DBP provided better estimation of invasive central SBP than brachial SBP/DBP calibration [50,51]. It was found that nocturnal SBP decline at the aorta and central arteries was significantly less pronounced than brachial SBP. Moreover, it was not related to a dipping of the heart rate and was virtually absent in the young individuals [52,53]. In our study, a nocturnal MAP decline <10% was diagnosed in 50.2% of the obese participants. The percentage of non-dippers was found to increase significantly, up to 83.3%, in patients with BMI ≥55 kg/m$^2$. The multivariable regression analysis revealed 3 groups of independent factors that reduced nocturnal MAP dipping: BW and BMI, reflecting body size; lean body mass, consisting mainly of muscle tissue; and AbdF/FM—the measure of central adiposity. It should be noted that abdominal fat in our study comprised almost the whole abdominal cavity and constituted 27.8% of total body fat—much more than the visceral adipose tissue measured in the aforementioned study [43]. Furthermore, it is important to note that a significant reduction of nocturnal MAP decline was observed only in participants in the highest quartiles of BW, BMI, LBM, and AbdF/F.

## Conclusion

The study found that the prevalence of hypertension in patients with obesity was as high as 66.7%, which value increased to 78.7% of the participants with BMI ≥50 kg/m$^2$. It was also found that hypertensive participants were characterized by larger measures of abdominal obesity, whereas normotensive individuals had higher FM, BF% and PerF. Furthermore, BW, BMI and LBM positively correlated with systolic BP and PP, while the measures of central adiposity positively correlated with SBP and DBP, but not with PP. A reduced nocturnal MAP decline was found in 50.2% of patients with obesity (BMI ≥30 kg/m$^2$) and in 83.3% of the participants with morbid obesity (BMI ≥55 kg/m$^2$). The multivariable analysis revealed that BW, BMI, LBM and AbdF/FM were independent factors that influenced MAP decline, and showed that the patients in the highest quartiles of BW, BMI, LBM and AbdF/FM had reduced nocturnal MAP dipping compared with the individuals in the lowest respective quartiles.

## Supporting information

**S1 Fig. The number and percentage of non-dippers among hypertensive patients by BMI values.**
(TIF)

**S2 Fig. The number and percentage of non-dippers among normotensive patients by BMI values.**
(TIF)

**S1 Table. The number and percentage (in brackets) of hypertensive and normotensive women and men.**
(DOCX)

**S2 Table. Anthropometric parameters and body composition in hypertensive and normo-tensive women with obesity.**
(DOCX)

**S3 Table. Anthropometric parameters and body composition in hypertensive and normo-tensive men with obesity.**
(DOCX)

## Author Contributions

**Conceptualization:** Marek Tałałaj.

**Data curation:** Agata Bogołowska-Stieblich, Ada Sawicka.

**Formal analysis:** Agata Bogołowska-Stieblich.

**Investigation:** Michał Wąsowski, Ada Sawicka.

**Supervision:** Piotr Jankowski.

**Writing – original draft:** Marek Tałałaj.

**Writing – review & editing:** Marek Tałałaj, Piotr Jankowski.

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
