## [Decision Letter · Decision Letter 0]

8 Aug 2022

PONE-D-22-12990The influence of body composition and fat distribution on circadian blood pressure rhythm and nocturnal mean arterial pressure dipping in patients with obesityPLOS ONE

Dear Dr. Tałałaj,

Thank you for submitting your manuscript to PLOS ONE. After careful consideration, we feel that it has merit but does not fully meet PLOS ONE’s publication criteria as it currently stands. Therefore, we invite you to submit a revised version of the manuscript that addresses the points raised during the review process.

 Reviewers found your article of interest but each suggested comments for improvement. In particular, make sure your rationale is clear and easy to follow in the introduction. There were also some concerns with the methodology, so please ensure that is addressed in the revision.

We look forward to receiving your revised manuscript.

Kind regards,

Jeremy P Loenneke

Academic Editor

PLOS ONE

Journal Requirements:

“The study was supported by the Grant Number 501-3-40-10-15, Centre of Postgraduate Medical Education, Warsaw, Poland”

Reviewers' comments:

Reviewer's Responses to Questions

**Comments to the Author**

1. Is the manuscript technically sound, and do the data support the conclusions?

Reviewer #1: Yes

Reviewer #2: Yes

2. Has the statistical analysis been performed appropriately and rigorously? 

Reviewer #1: Yes

Reviewer #2: Yes

3. Have the authors made all data underlying the findings in their manuscript fully available?

Reviewer #1: Yes

Reviewer #2: Yes

4. Is the manuscript presented in an intelligible fashion and written in standard English?

Reviewer #1: No

Reviewer #2: Yes

5. Review Comments to the Author

Reviewer #1: General Comments:

This paper explores an interesting question, however, some significant rewriting of the article for clarity and correction of grammatical error is required. Also, there are some issues with the paper described below.

It would’ve been interesting to explore further how hypertension status within BMI group relates to the nocturnal blood pressure dipping status (even though non-dipping is an independent predictor of CV risk and mortality, as stated by the authors.)

Introduction

The rationale built here is difficult to follow. Particularly, the section on central blood pressure does not fit in. This paper does not measure central blood pressure.

Methods

generally sound

Results

-only report results to the decimal to which they were measured (rules of significant figures)

-line 136 and 137 – “blood pressure lowering drugs, 1-6” To what is this referring?

-Tables 2 and 6 – the “(No)” is confusing; please find a way to present this differently

-Table 3 - as men and women should statistically significant differences in systolic and diastolic pressure that day, night, and in the mean value, it would strengthen the paper to provide further analysis examining variables with men alone and with women alone. It is interesting to note that the females alone would not even reach the hypertension classification while the males would. If this kind of analysis were added, in my opinion, some of the other analyses performed could be eliminated so as not to bloat the paper. Keep the analyses simple and related to the main purpose of the paper.

Discussion

-Line 311 – This is interesting: “whereas normotensive individuals had larger FM, BF%, and PerF.” the authors mention this finding in passing (line 250, “a possible protective role for peripheral adipose tissue” but it is not really discussed.

-line 293 - if central blood pressure measurement is arguably better, and “nocturnal SPB decline of the aorta and central arteries was significantly less pronounced compared with brachial SBP” (refs 42,43), how important are the peripheral nocturnal blood pressure measurements? What was the relationship between central dipping and peripheral dipping in other studies?

Reviewer #2: Thank you for the opportunity to review this manuscript. The authors sought to investigate the relationship between markers of body composition and the distribution of fat on nocturnal blood pressure dipping. The results suggest that those with the highest body weight and, BMI, and lean body mass, had reduced nocturnal blood pressure dipper. The manuscript is very interesting; however, improvement is required in some areas before it is ready for publication.

6. PLOS authors have the option to publish the peer review history of their article (what does this mean?). If published, this will include your full peer review and any attached files.

Reviewer #1: No

Reviewer #2: No

---

## [Author Response · Author response to Decision Letter 0]

25 Sep 2022

Response to reviewers

Reviewer #1: 

General Comments:

This paper explores an interesting question, however, some significant rewriting of the article for clarity and correction of grammatical error is required. 

The whole article has been changed and corrected.

It would’ve been interesting to explore further how hypertension status within BMI group relates to the nocturnal blood pressure dipping status (even though non-dipping is an independent predictor of CV risk and mortality, as stated by the authors.)

As the number of normotensive patients in the study is low, the additional analysis is shown as supporting data. 

Introduction

The rationale built here is difficult to follow. Particularly, the section on central blood pressure does not fit in. This paper does not measure central blood pressure.

The introduction has been extended and corrected to make the rationale clearer and easy to follow.

Results

-only report results to the decimal to which they were measured (rules of significant figures)

The results have been corrected according to the suggestion.

-line 136 and 137 – “blood pressure lowering drugs, 1-6” To what is this referring?

The sentence has been changed to make it clearer.

-Tables 2 and 6 – the “(No)” is confusing; please find a way to present this differently

‘No’ was replaced by ‘n’.

-Table 3 - as men and women should statistically significant differences in systolic and diastolic pressure that day, night, and in the mean value, it would strengthen the paper to provide further analysis examining variables with men alone and with women alone. It is interesting to note that the females alone would not even reach the hypertension classification while the males would. If this kind of analysis were added, in my opinion, some of the other analyses performed could be eliminated so as not to bloat the paper. Keep the analyses simple and related to the main purpose of the paper.

Extended analysis has been included in the supporting information files. 

Discussion

-Line 311 – This is interesting: “whereas normotensive individuals had larger FM, BF%, and PerF.” the authors mention this finding in passing (line 250, “a possible protective role for peripheral adipose tissue” but it is not really discussed.

-line 293 - if central blood pressure measurement is arguably better, and “nocturnal SPB decline of the aorta and central arteries was significantly less pronounced compared with brachial SBP” (refs 42,43), how important are the peripheral nocturnal blood pressure measurements? What was the relationship between central dipping and peripheral dipping in other studies?

The discussion on the role of subcutaneous fat has been expanded significantly. 

The role of peripheral nocturnal blood pressure measurements, and the relationship between central dipping and peripheral dipping, are additionally discussed. 

Reviewer #2

Introduction

Line 54-55 “As the best currently available method of measuring blood pressure (BP), and its circadian profile, has been regarded 24-hour ambulatory blood pressure monitoring (ABPM).”

 Please consider changing this line to read “24-hour ambulatory blood pressure monitoring (ABPM) has been regarded as the best currently available method of measuring blood pressure (BP), and its circadian profile.” 

Additionally, who has made this claim please support this with a citation. 

The sentence has been changed according to the suggestion. A citation has been added.

Methods

Lines 84-86 “Waist circumference was determined in a horizontal plane, midway between the iliac crest and the costal margin, at the end of a normal expiration, using a non stretch tape measure.”

Which organization recommends taking it this way? This is different than how the American College of Sports Medicine describes the waist measurement. Please clarify why it was measured this way. Additionally, how was the hip measurement determined. 

There are different waist circumference measurement protocols, and there is no consensus. 

The measurement ‘midway between the iliac crest and the costal margin’ is recommended by the WHO, ASTM International and ISO 7250-1:2017, among others. By taking the measurement ‘midway between the iliac crest and the costal margin’, we wanted to obtain better reliability due to stable landmarks and better correlation with visceral adipose tissue mass (the abdomen in people with obesity is more in the shape of a ball than an hour-glass).

Hip circumference was measured at the level of the femoral great trochanter.

Piqueras P et al. Anthropometric indicators as a tool for diagnosis of obesity and other health risk factors: a literature review. Front Psychol 2021;12:631179. doi: 10.3389/fpsyg.2021.631179 

Serviente C, Sforzo GA. A simple yet complicated tool measuring waist circumference to determine cardiometabolic risk. ACSM’s Health & Fitness Journal 2013; 17: 29-34.

doi: 10.1249/FIT.0b013e3182a956f5

Lines 97-99 “Systolic and diastolic BP were measured every 30 minutes at daytime (between 6.00 and 22.00) and every 60 minutes at night (from 22.00 to 6.00). Average daytime and nighttime BP were computed as the means of all readings during each period.”

How was mean blood pressure determined? There are 32 daytime measurements and 8 nighttime measurements. This may be problematic as this unfairly weights the influence of day time blood pressure. Honestly, I am not sure the best way to do this. But the current method is problematic. Either way please clarify how you calculated mean blood pressure. 

I have to agree that ‘this unfairly weights the influence of daytime blood pressure’. The mean 24-hour values of SBP and DBP presented in the paper were 127.8 and 75.4 mmHg, respectively. Assuming that all 40 measurements (32 + 8) are legible, mean SBP and DBP should be 128.5 and 75.9 mmHg, respectively (daytime/nighttime measurements in the proportion of 4 to 1). If we calculate mean 24-hour SBP and DBP values including diurnal and nocturnal measurements in a 2-to-1 ratio (16 + 8 hours), the average SBP and DBP should be 127.2 and 74.8 mmHg, respectively. This means that approximately 1/3 of the daytime readings were not legible, and that the real daytime-to-nighttime measurement ratio was approximately 3 to 1. 

Unfortunately, the exact numbers of legible measurements were not collected in our database, and are contained in individual patients’ source data only.

In the paper we decided to write that ‘the average circadian systolic and diastolic BP were calculated as the means of all legible measurements within a 24-hour period.’

Results.

Lines 184-185 “The assessment of nocturnal BP decline revealed that the magnitude of DBP dipping (10.8 +/- 8.7%) was significantly greater than SBP decline (7.6 +/- 6.8%, p<0.001).”

How was this determined? It is also more informative to the reader if you put the difference 

Between SBP and DBP. Please clarify the percentages listed. It was stated in the statistical analyses section that it is standard deviations. That does not seem to be what is presented here. 

The assessment of nocturnal BP decline revealed that the magnitude of SBP dipping (9.9 ± 8.2 mmHg) was significantly greater than DBP decline (8.5 ± 7.0 mmHg, p<0.001), while the percentage decline of DBP (10.8 ± 8.7%) was greater than SBP decline (7.6 ± 6.8%, p<0.001). 

Discussion 

Lines 300-301 “It should be noted, that abdominal fat determined in our study comprised almost whole abdominal cavity and constituted 27,8% of”. That comma should be a period. 

The proper correction was made

---

## [Decision Letter · Decision Letter 1]

19 Oct 2022

PONE-D-22-12990R1The influence of body composition and fat distribution on circadian blood pressure rhythm and nocturnal mean arterial pressure dipping in patients with obesityPLOS ONE

Dear Dr. Tałałaj,

Thank you for submitting your manuscript to PLOS ONE. After careful consideration, we feel that it has merit but does not fully meet PLOS ONE’s publication criteria as it currently stands. Therefore, we invite you to submit a revised version of the manuscript that addresses the points raised during the review process.

Both reviewers found that the manuscript was improved, however, each had additional minor comments that need to be addressed.

We look forward to receiving your revised manuscript.

Kind regards,

Jeremy P Loenneke

Academic Editor

PLOS ONE

Journal Requirements:

Reviewers' comments:

Reviewer's Responses to Questions

**Comments to the Author**

1. If the authors have adequately addressed your comments raised in a previous round of review and you feel that this manuscript is now acceptable for publication, you may indicate that here to bypass the “Comments to the Author” section, enter your conflict of interest statement in the “Confidential to Editor” section, and submit your "Accept" recommendation.

Reviewer #1: (No Response)

Reviewer #2: (No Response)

2. Is the manuscript technically sound, and do the data support the conclusions?

Reviewer #1: Yes

Reviewer #2: Yes

3. Has the statistical analysis been performed appropriately and rigorously? 

Reviewer #1: Yes

Reviewer #2: Yes

4. Have the authors made all data underlying the findings in their manuscript fully available?

Reviewer #1: Yes

Reviewer #2: Yes

5. Is the manuscript presented in an intelligible fashion and written in standard English?

Reviewer #1: Yes

Reviewer #2: Yes

6. Review Comments to the Author

Reviewer #1: Overall, this paper is significantly improved. It addresses an important question is generally clearly presented. The grammar and punctuation issues have clearly been addressed although some minor grammar and punctuation issues remain.

Abstract

• Still some minor grammatical correct / punctuation issues

Intro

• Significantly improved! - although still some minor grammatical correct / punctuation issues

Results

• decimals are still shown for values for which it was unlikely the original measurement was to a decimal (age and BP, for example)

• “Blood pressure-lowering drugs, 1 137 through 6, have been taken by 289 participants” � where can the reader find the 1-6 info?

Discussion

• Still some minor grammatical correct / punctuation issues – review your comma usage particularly (while comma usage can be subjective, there are a few cases throughout the paper where the usage is clearly incorrect. For example, “Previous studies 288 have documented, that mean arterial pressure better reflected central, aortic SBP, and better 289 predicted cardiovascular risk, as compared with brachial SBP [38,39].” Between “documented” and “that” there clearly should not be a comma. This type of error occurs in other places.

Reviewer #2: Thank you for the opportunity to re-review this manuscript. The authors addressed most of my comments. I have still have three points that need to be addressed.

Page 5 lines 90-94

“Waist and hip circumferences (WC, HC) were measured to the nearest 1 cm, and the waist-to-hip ratio (WHR) was calculated. WC was determined in a horizontal plane, midway between the iliac crest and the costal margin, at the end of a normal expiration; HC was measured at the level of the greater trochanter, using a non-stretch tape measure.”

Sorry if my last comment on this was not clear. I agree that there is no consensus on how to measure waist circumference. I am just suggesting that the authors provide a reference for who recommends this method of measurement.

Page 6 lines 105-112

“Systolic and diastolic BP (DBP) were measured every 30 minutes during the day (between 6.00 and 22.00) and every 60 minutes during the night (from 22.00 to 6.00). The average daytime and nighttime BP were computed as the means of all readings during each period, and the average circadian SBP and DBP were calculated as the means of all legible measurements within a 24-hour period. Hypertension was diagnosed according to the 2018 European Society of Cardiology guidelines [7]. Non-dipping of BP was defined as a decline in MAP values <10% from the average daytime to the average nighttime values.

“I have to agree that ‘this unfairly weights the influence of daytime blood pressure’. The mean 24-hour values of SBP and DBP presented in the paper were 127.8 and 75.4 mmHg, respectively. Assuming that all 40 measurements (32 + 8) are legible, mean SBP and DBP should be 128.5 and 75.9 mmHg, respectively (daytime/nighttime measurements in the proportion of 4 to 1). If we calculate mean 24-hour SBP and DBP values including diurnal and nocturnal measurements in a 2-to-1 ratio (16 + 8 hours), the average SBP and DBP should be 127.2 and 74.8 mmHg, respectively. This means that approximately 1/3 of the daytime readings were not legible, and that the real daytime-to-nighttime measurement ratio was approximately 3 to 1.

Unfortunately, the exact numbers of legible measurements were not collected in our database, and are contained in individual patients’ source data only.

In the paper we decided to write that ‘the average circadian systolic and diastolic BP were calculated as the means of all legible measurements within a 24-hour period.”

I appreciate the authors honesty. But they need to explicitly state this in the manuscript. It is important the reader knows how the data was affected by the unequal number of measurements.

Page 12 lines 194-198

The assessment of nocturnal BP decline revealed that the magnitude of SBP dipping (9.9 ± 8.2

mmHg) was significantly greater than that of the DBP dipping (8.5 ± 7.0 mmHg, p<0.001), while the percentage of the DBP decline (10.8 ± 8.7%) was greater than that of SBP decline (7.6 ± 6.8%, p<0.001). The average MAP dipping was 9.4 ± 7.5%. Reduced nocturnal MAP decline was diagnosed in 219 patients (50.2%).

The authors should consider reported the difference and SD of that difference between the SBP and DBP dipping. That is the information we are interested in.

7. PLOS authors have the option to publish the peer review history of their article (what does this mean?). If published, this will include your full peer review and any attached files.

Reviewer #1: No

Reviewer #2: No

---

## [Author Response · Author response to Decision Letter 1]

20 Nov 2022

Response to reviewers

Reviewer #1: 

Abstract

• Still some minor grammatical correct / punctuation issues

Intro

• Significantly improved! - although still some minor grammatical correct / punctuation issues

Discussion

The grammar and punctuation issues have clearly been addressed although some minor grammar and punctuation issues remain. Review your comma usage particularly (while comma usage can be subjective, there are a few cases throughout the paper where the usage is clearly incorrect. For example, “Previous studies 288 have documented, that mean arterial pressure better reflected central, aortic SBP, and better 289 predicted cardiovascular risk, as compared with brachial SBP [38,39].” Between “documented” and “that” there clearly should not be a comma. This type of error occurs in other places.

The text has been corrected. We believe no more grammar and punctuation issues remained.

Results

• decimals are still shown for values for which it was unlikely the original measurement was to a decimal (age and BP, for example)

Decimals were removed. The only exception can be found on page 12, lines 198-199, as the differences between mean BP values are very small. 

• “Blood pressure-lowering drugs, 1 137 through 6, have been taken by 289 participants” where can the reader find the 1-6 info?

The sentence has been changed to make it more clear (lines 150-151).

Reviewer #2: 

Thank you for the opportunity to re-review this manuscript. The authors addressed most of my comments. I have still have three points that need to be addressed.

Page 5 lines 90-94

“Waist and hip circumferences (WC, HC) were measured to the nearest 1 cm, and the waist-to-hip ratio (WHR) was calculated. WC was determined in a horizontal plane, midway between the iliac crest and the costal margin, at the end of a normal expiration; HC was measured at the level of the greater trochanter, using a non-stretch tape measure.”

Sorry if my last comment on this was not clear. I agree that there is no consensus on how to measure waist circumference. I am just suggesting that the authors provide a reference for who recommends this method of measurement.

A reference has been provided. Line 94 [19]. 

Page 6 lines 105-112

“Systolic and diastolic BP (DBP) were measured every 30 minutes during the day (between 6.00 and 22.00) and every 60 minutes during the night (from 22.00 to 6.00). The average daytime and nighttime BP were computed as the means of all readings during each period, and the average circadian SBP and DBP were calculated as the means of all legible measurements within a 24-hour period. Hypertension was diagnosed according to the 2018 European Society of Cardiology guidelines [7]. Non-dipping of BP was defined as a decline in MAP values <10% from the average daytime to the average nighttime values.

“I have to agree that ‘this unfairly weights the influence of daytime blood pressure’. The mean 24-hour values of SBP and DBP presented in the paper were 127.8 and 75.4 mmHg, respectively. Assuming that all 40 measurements (32 + 8) are legible, mean SBP and DBP should be 128.5 and 75.9 mmHg, respectively (daytime/nighttime measurements in the proportion of 4 to 1). If we calculate mean 24-hour SBP and DBP values including diurnal and nocturnal measurements in a 2-to-1 ratio (16 + 8 hours), the average SBP and DBP should be 127.2 and 74.8 mmHg, respectively. This means that approximately 1/3 of the daytime readings were not legible, and that the real daytime-to-nighttime measurement ratio was approximately 3 to 1.

Unfortunately, the exact numbers of legible measurements were not collected in our database, and are contained in individual patients’ source data only.

In the paper we decided to write that ‘the average circadian systolic and diastolic BP were calculated as the means of all legible measurements within a 24-hour period.”

I appreciate the authors honesty. But they need to explicitly state this in the manuscript. It is important the reader knows how the data was affected by the unequal number of measurements.

The extended information on BP measurements was added: page 8, lines 144-147. 

Page 12 lines 194-198

The assessment of nocturnal BP decline revealed that the magnitude of SBP dipping (9.9 ± 8.2 mmHg) was significantly greater than that of the DBP dipping (8.5 ± 7.0 mmHg, p<0.001), while the percentage of the DBP decline (10.8 ± 8.7%) was greater than that of SBP decline (7.6 ± 6.8%, p<0.001). The average MAP dipping was 9.4 ± 7.5%. Reduced nocturnal MAP decline was diagnosed in 219 patients (50.2%).

The authors should consider reported the difference and SD of that difference between the SBP and DBP dipping. That is the information we are interested in.

The data on the difference and SD of that difference between the SBP and DBP dipping has been added: page 12, lines 198-201.

---

## [Decision Letter · Decision Letter 2]

14 Dec 2022

PONE-D-22-12990R2The influence of body composition and fat distribution on circadian blood pressure rhythm and nocturnal mean arterial pressure dipping in patients with obesityPLOS ONE

Dear Dr. Tałałaj,

Thank you for submitting your manuscript to PLOS ONE. After careful consideration, we feel that it has merit but does not fully meet PLOS ONE’s publication criteria as it currently stands. Therefore, we invite you to submit a revised version of the manuscript that addresses the points raised during the review process.

 A reviewer has requested additional minor revisions in order to add clarity related to the ratio of measurements used.

We look forward to receiving your revised manuscript.

Kind regards,

Jeremy P Loenneke

Academic Editor

PLOS ONE

Journal Requirements:

Reviewers' comments:

Reviewer's Responses to Questions

**Comments to the Author**

1. If the authors have adequately addressed your comments raised in a previous round of review and you feel that this manuscript is now acceptable for publication, you may indicate that here to bypass the “Comments to the Author” section, enter your conflict of interest statement in the “Confidential to Editor” section, and submit your "Accept" recommendation.

Reviewer #1: All comments have been addressed

Reviewer #2: (No Response)

2. Is the manuscript technically sound, and do the data support the conclusions?

Reviewer #1: Yes

Reviewer #2: Yes

3. Has the statistical analysis been performed appropriately and rigorously? 

Reviewer #1: Yes

Reviewer #2: Yes

4. Have the authors made all data underlying the findings in their manuscript fully available?

Reviewer #1: No

Reviewer #2: Yes

5. Is the manuscript presented in an intelligible fashion and written in standard English?

Reviewer #1: Yes

Reviewer #2: No

6. Review Comments to the Author

Reviewer #1: All comments have been addressed. I have no further comments. I appreciate the work the authors did on this manuscript.

Reviewer #2: Thank you again for the opportunity to review this manuscript. The authors have made substantial improvements to the paper. However, I still have one concern that should be addressed.

Page 8 lines 144-147.

“The 24-hour BP monitoring included 40 measurements: 32 daytime and 8 nighttime readings. The comparison of average daytime, average nighttime and mean diurnal SBP and DBP values shows that actual daytime-to-nighttime ratio of legible BP measurements was 3 to 1. That means that approximately 1/3 of the daytime readings were not legible.”

I appreciate the authors adding this section in. However, I believe more discussion is warranted about why this is a problem. Perhaps discussing this in the limitations could be useful. But ultimately, I think the authors needs to discuss why the ratio is important. Simply when looking at mean blood pressure the data is skewed to be more like the day time measurements compared to the night time measurements due to the increased number of measurements taken during the day.

7. PLOS authors have the option to publish the peer review history of their article (what does this mean?). If published, this will include your full peer review and any attached files.

Reviewer #1: No

Reviewer #2: No

---

## [Author Response · Author response to Decision Letter 2]

8 Jan 2023

Response to reviewers

Reviewer #1: 

All comments have been addressed. I have no further comments. I appreciate the work the authors did on this manuscript.

Thank you for your opinion

Reviewer #2: 

The authors have made substantial improvements to the paper. However, I still have one concern that should be addressed.

Page 8 lines 144-147.

“The 24-hour BP monitoring included 40 measurements: 32 daytime and 8 nighttime readings. The comparison of average daytime, average nighttime and mean diurnal SBP and DBP values shows that actual daytime-to-nighttime ratio of legible BP measurements was 3 to 1. That means that approximately 1/3 of the daytime readings were not legible.”

I appreciate the authors adding this section in. However, I believe more discussion is warranted about why this is a problem. Perhaps discussing this in the limitations could be useful. But ultimately, I think the authors needs to discuss why the ratio is important. Simply when looking at mean blood pressure the data is skewed to be more like the day time measurements compared to the night time measurements due to the increased number of measurements taken during the day.

To solve the problem the average circadian SBP and DBP were calculated according to the formula (2 x mean daytime BP + mean nighttime BP) / 3. 

This formula takes into account the times of diurnal and nocturnal BP measurements in a ratio of 2-to-1 (16-to-8 hours) and eliminates the influence of different measurement frequencies on average 24-hour SBP and DBP values. 

As a consequence, the text on page 8, lines 145-148, became redundant and could have been removed. 

The new calculations minimally changed the figures in tables 1, 3 and 5 but did not affect the results.

---

## [Decision Letter · Decision Letter 3]

16 Jan 2023

The influence of body composition and fat distribution on circadian blood pressure rhythm and nocturnal mean arterial pressure dipping in patients with obesity

PONE-D-22-12990R3

Dear Dr. Tałałaj,

We’re pleased to inform you that your manuscript has been judged scientifically suitable for publication and will be formally accepted for publication once it meets all outstanding technical requirements.

Kind regards,

Jeremy P Loenneke

Academic Editor

PLOS ONE

Additional Editor Comments (optional):

Reviewers' comments:

Reviewer's Responses to Questions

**Comments to the Author**

1. If the authors have adequately addressed your comments raised in a previous round of review and you feel that this manuscript is now acceptable for publication, you may indicate that here to bypass the “Comments to the Author” section, enter your conflict of interest statement in the “Confidential to Editor” section, and submit your "Accept" recommendation.

Reviewer #2: All comments have been addressed

2. Is the manuscript technically sound, and do the data support the conclusions?

Reviewer #2: Yes

3. Has the statistical analysis been performed appropriately and rigorously? 

Reviewer #2: Yes

4. Have the authors made all data underlying the findings in their manuscript fully available?

Reviewer #2: Yes

5. Is the manuscript presented in an intelligible fashion and written in standard English?

Reviewer #2: Yes

6. Review Comments to the Author

Reviewer #2: The authors have made substantial improvements to the paper. I appreciate the authors hard work in preparing this manuscript

7. PLOS authors have the option to publish the peer review history of their article (what does this mean?). If published, this will include your full peer review and any attached files.

Reviewer #2: No

---

## [Editor Report · Acceptance letter]

23 Jan 2023

PONE-D-22-12990R3 

The influence of body composition and fat distribution on circadian blood pressure rhythm and nocturnal mean arterial pressure dipping in patients with obesity 

Dear Dr. Tałałaj:

I'm pleased to inform you that your manuscript has been deemed suitable for publication in PLOS ONE. Congratulations! Your manuscript is now with our production department. 

Kind regards, 

on behalf of

Dr. Jeremy P Loenneke 

Academic Editor

PLOS ONE